# Identifying the significant drivers of containerized freight rates: From the perspective of dynamic multiscale dependence

Yanhui Chen[1]*, Ailing Feng[2], Chao Tang[3]

1 School of Economics and Management, Shanghai Maritime University, Shanghai, China, 2 College of Economics and Management, Nanjing University of Aeronautics and Astronautics, Nanjing, China, 3 The 9th Shanghai Pudong New District Youth Federation Committee, Shanghai, China

* chen.yan-hui@my.cityu.edu.hk

## Abstract

In August 2023, the launch of Shanghai Containerized Freight Index (SCFI) futures provides a suitable tool for risk management in the container shipping market, as well as new options for risk management of other financial assets. However, limited research exists on the influencing factors behind container freight rate fluctuations. This paper explores the nonlinear dynamic interdependence between the SCFI and 12 factors from the stock, commodity, carbon, and other markets using a data decomposition–reconstruction-based time-varying copula method, which can assist the stakeholders in hedging risk at different timescales. The findings reveal that most factors show no or limited upper tail dependence with SCFI in the short term. Medium- and long-term dependence is significantly stronger, indicating structural connections over longer horizons. Moreover, the dependence intensifies during extreme risk events. Generally, downside tail risks exert a greater influence on SCFI in the medium to long term, while upside tail risks are found to affect SCFI at any time horizon. This paper focuses on the tail risk interdependence analysis between SCFI and other assets, because the launch of SCFI futures makes the stakeholders to use this future to build risk management portfolios with other assets inevitably. The result provides useful implications to stakeholders with varying financial or investment attributes associated with shipping industry, aiding them in clarifying the different tail risk associations between SCFI futures and other assets at different timescales.

## Introduction

Container shipping is an essential way in facilitating cost-effective and efficient cross-border goods transportation, thereby serving as a vital component of the global economy [1]. Freight rates, as a critical element within the shipping market, play a pivotal role in adjusting the demand and supply dynamics in the shipping market. To meet the needs of developing international container freight index derivatives

**Data availability statement:** All relevant data are within the manuscript and its Supporting Information files.

**Funding:** This research was funded by the National Social Science Foundation of China (Grant numbers 25BJY044). The funders did not play any role in the study design, data collection and analysis, decision to publish, or preparation of the manuscript.

**Competing interests:** Dr. Chao Tang is hired by Shanghai Futures Exchange but this study does not represent the opinions of the Shanghai Futures Exchange. This does not alter our adherence to PLOS ONE policies on sharing data and materials.

and improving China's export container freight index system, the Shanghai (Export) Containerized Freight Index (SCFI) was officially released on October 16, 2009. Subsequently, in 2020, the Shanghai Shipping Exchange (SSE) published the Shanghai Containerized Settlement Freight Index (SCFIS), specifically for freight rate derivatives. In August 2023, SCFIS (Europe Service) futures were first listed on the Shanghai International Energy Exchange. SCFIS (Europe Service) futures are derivative instruments available to global investors on an international scale. According to SSE, SCFIS has the same trend as the SCFI, as its correlation with SCFI has reached 98%. Because the historical data of SCFI is richer than that of SCFIS, and it can also better reflect the comprehensive situation of the container shipping market, this paper selects SCFI as a proxy variable of SCFIS (Europe Service), the underlying index of the newly listed freight rates futures, for empirical research. The rising demand for risk management in container transportation is driven by internal and external environmental factors. Analyzing the tail dependence between SCFI and key drivers offers crucial insights for effective risk management in the industry.

Gaining a comprehensive understanding of the mechanism and dynamics behind the determination of shipping freight rates holds meaningful implications for industry and academia involved in maritime transport [2]. Due to the seasonality, non-stationary and nonlinear nature of freight rates and the complexity of influencing factors, analyzing the fluctuations of freight rates in the shipping market is a highly challenging job [1,3]. Therefore, a massive research literature has been inspired to explore the drivers of shipping freight rates ever since the early research of Alizadeh and Muradoglu [4]. The prevailing view among researchers suggests that the shipping market is subject to notable influences from fluctuations in the dynamics of supply and demand [5–7]. Researchers have emphasized the relationship between the shipping market and primary commodity markets [4,7–11], the mutual feedback between stock and maritime markets, and the impact of risk events in understanding fluctuations in the shipping market [12–15]. Additionally, a recent research by Meng et al. [16] suggested a strong temporal and frequency dependence between the carbon finance market and the shipping market, specifically, the carbon market exerting a leading influence. Li et al. [17] pointed out that there is a one-way causal relationship from the dry bulk and tanker markets to the container shipping market.

In addition to analyzing the fluctuation characteristics and economic drivers of shipping freight rates, research on shipping derivatives (forward freight agreements) has also occasionally emerged in the past 15 years. Sel and Minner [18] proposed a data-driven hedging policy for Forward Freight Agreements (FFAs) based on comparisons of FFA rates with future spot rate forecasts. Gong et al. [19] designed portfolios consisting of diverse quarterly FFAs to maximize the market participants' expected utility. Alizadeh [20] explored the price volatility and trading volume relationship in the FFA market for dry bulk ships. Gong and Lu [21] studied the spillover effect of spot freight rates on FFAs in the Capesize market. Benth and Koekebakker [22] used stochastic models to derive theoretical Supramax FFA rates. Gu et al. [23] compared the hedging performance of FFAs in the Panamax and Capesize sectors. Sun et al. [24] investigated the dynamic impacts of skewness on the risk-return relationship among

dry bulk spot freight rates and the corresponding FFAs. Shi et al. [25] explored the minimum Value-at-Risk hedge ratios with spot freight rates and FFAs in the tanker market. All the research subjects are FFAs associated with the dry bulk and tanker markets. There is a lack of research related to container freight rate derivatives. In addition, most studies only focus on the nexus between the spot market and FFAs, such as Gong and Lu [21] and Sun et al. [24]. This article is motivated by exploring the potential risk linkage between container freight rate futures and other assets, which intends to assist some financial institutions in using SCFI futures for risk management of other assets.

While numerous researchers have provided valuable insights into the driving mechanism of freight rates, it is important to acknowledge that limitations exist in our current understanding of their formation. Firstly, previous scholarly investigations primarily emphasize the dry bulk and tanker seaborne transport, with limited concentration given to the container segment [2,26]. Jeon et al. [27] pointed out that the container shipping sector possesses distinct characteristics compared to other segments of the shipping industry. Existing literature has predominantly focused on forecasting the containerized freight index [28–33]. Comprehensive academic studies that specifically address the impact factors on container shipping rates are relatively scarce.

Secondly, the existing researches predominantly focus on static and dynamic analysis of freight rates at the overall level, while overlooking potential driving mechanisms at different time scales. The shipping market comprises various investors with diverse financial characteristics, engaging in activities that span a wide range of time scales, including speculative behaviors that occur within minutes and investment behaviors that extend over several years [34]. Multiscale analysis has been widely used in analyzing the internal and latent data-generating process in both time and frequency domains [35,36]. More recently, maritime scholars have adopted multiscale copula frameworks to study the interdependence structure among shipping market variables and related economic and financial variables. Zhang [37] proposed a multiscale copula approach based on Seasonal and Trend decomposition (STL) to examine the interdependencies among oil price, bunker fuel price, and freight rate index. To identify the relationship between the studied indices in different time horizons, Bai [13] employed a wavelet-based multiscale copula model. However, aforementioned multiscale decomposition methods are only applicable to univariate analysis. In modeling the multivariate data, they decompose the multivariate data separately, which may ignore the internal relationship among the variables and lead to distortion [38]. When the dependence of multiple variables in different time domains is compared, the above method cannot ensure the consistency of time scales. For freight rate drivers, there are many influencing factors, such as stock price, crude oil price, political risk, etc. For this multivariable problem, the multiscale driving mechanism cannot be defined.

To overcome the deficiencies above, this paper concentrates on the container shipping market and purposes to explore and identify the nonlinear dynamic dependencies between SCFI and significant drivers at different timescales. A MEMD-based copula analysis framework is implemented to depict the multiscale, asymmetric, and tail dependencies among the variables. Specifically, the first step is to decompose the raw return series with the MEMD method and recombine them based on the permutation entropy and time horizons to obtain three short-, medium-, and long-term return series and a residual sequence set. Due to the complex data-generating process of container freight rates, many influencing factors need to be considered. Consequently, the second step is to analyze the short-term, medium-, and long-term factors through the least absolute shrinkage and selection operator (LASSO). The third step examines the dependence among the freight rates and the selected impact factors in a further step. The ARMA or ARMA-EGARCH models are established based on the return series at different timescales to depict the marginal distributions. Last but not least, the static and time-varying copula methods are used, respectively, to identify the different tail dependence styles among the selected impact factors and freight rates in different periods.

The contributions of this study are as follows. First of all, this study is a preliminary attempt to explore the tail-dependence among SCFI and its significant drivers in diversified market conditions with both static and dynamic analysis. To our knowledge, this study considers the most extensive factors on container freight rates, which can enrich the literature. Practically, this article can guide the traders to pay attention to different risk sources in different time dimensions.

Secondly, a research framework based on multiscale analysis and a copula model for exploring the freight rate drivers is proposed. The using of MEMD in multiscale analysis contributes to depict the internal interactions among different variables in time domain uniformly. Also, it is first used for multivariate shipping market analysis. The proposed multiscale copula model, which considers dynamic analysis, captures SCFI drivers hidden over different timescales, and thus helps to understand their driving mechanisms and economic implications across different time domains. Finally, it is worth mentioning that our results provide useful policy implications and applicable strategic planning to the relevant stakeholders with different trading cycles in container shipping and financial markets, since this paper analyzes the tail-dependence at different timescales both statically and dynamically. The understanding of the dynamic relationships can assist the participants in thinking ahead of the potential risks, and assist the policymakers in organizing a well-regulated market.

Our empirical results can be summarized in the following three points. Our observations suggest that short-term SCFI is influenced by factors that have either no or limited upper tail dependence with SCFI, implying that upper extreme events have a notable effect on the SCFI in the short term. Secondly, significantly greater dependence and more interdependent structure between SCFI and impact factors in the medium- and long-term. The results from the medium-term time-varying copula analysis indicate that the dependence during extreme risk events is stronger compared to normal periods, particularly in the years 2012, 2016, 2020, 2022, and 2024. Consequently, lower tail extreme events in relevant markets have an important impact on SCFI over both the medium- and long-term, while upper extreme events can exert a substantial influence on SCFI through the entire duration.

The following parts of this study includes: Section 2 which specifies the main methods and the research framework; Section 3 which presents the data and descriptive statistics; Section 4 which reports and analyzes the results; and Section 5 which concludes the paper and points out the implications.

## Methodology

### Multivariate empirical mode decomposition (MEMD)

MEMD (Multivariate Empirical Mode Decomposition) is an extension of EMD. The key superiority of this method rests with its ability to simultaneously decompose multivariate time series data into different frequency domain data self-adaptively, which ensures an equal number of intrinsic mode functions (IMFs). Moreover, these IMFs are effectively aligned based on a frequency scale, facilitating a matched composition of IMF components in terms of both quantity and scale. This research analyzes the risk dependence between different variables and SCFI at different time scales. In order to make the relationship between each variable and SCFI comparable, it is necessary to standardize the time scale. MEMD can decompose multiple variables within a unified framework, which perfectly solves this problem. Rehman and Mandic [39] proposed the basic steps of MEMD, which are as follows.

(1) Select a suitable point set on the $n-1$ sphere and sample it.

(2) For all $k$ groups (the entire set of direction vectors), the set $\{p^{\theta_k}(t)\}_{k=1}^{K}$ is used as the projection set to calculate the input signal $\{V(t)\}_{t=1}^{T}$ along the direction vector $X^{\theta_k}$, where $X^{\theta_k} = \{x_1^k, x_2^k, \ldots, x_n^k\}$ and $\{V(t)\}_{t=1}^{T} = \{v_1(t), v_2(t), \ldots, v_n(t)\}$.

(3) Find the corresponding projection signal $\{p^{\theta_k}(t)\}_{k=1}^{K}$ maximum time point $\{t_i^{\theta_k}\}$.

(4) For $[t_i^{\theta_k}, V(t_i^{\theta_k})]$ to obtain the multivariate envelope curve $\{e^{\theta_k}(t)\}_{k=1}^{K}$.

(5) For a group of direction vectors $K$, calculate the average value $m(T)$ of the envelope curve, and its calculation formula is given as:

$$m(t) = \frac{1}{K} \sum_{k=1}^{K} e^{\theta_k}(t).$$

(1)

(6) Find $d(t)$ by $d(t) = x(t) - m(t)$. If $d(t)$ meets the stop criteria of multiple IMF, it will be regarded as an intrinsic mode function. Otherwise, return to Step 2. Among them, the stop criterion is that after continuous multiple screening, the mean value of the upper envelope and the lower envelope must be approximately equal to zero or meet some requirements.

Finally, the original time series can be decomposed into several IMF components and one residual, which can be written as:

$$x(t) = \sum_{i=1}^{n} IMF_i(t) + Res(t). \tag{2}$$

Where $n$ is the number of IMFs, $x(t)$ indicates one of the time series in the multivariate data.

**Least absolute shrinkage and selection operator (LASSO)**

LASSO was first proposed by Tibshirani [40]. Its essence is the least squares estimation with penalty factor, which compresses the coefficient with small absolute value to 0 through the penalty term, so as to achieve the purpose of variable selection and parameter estimation. The LASSO method has been found to effectively identify influential variables and those that can offer additional valuable information [41]. LASSO achieves model sparsity through $L_1$ regularization, thereby improving the interpretability and generalization ability of the model. It has advantages in multivariate selection and is particularly suitable when it is necessary to clarify the importance of variables. This method helps us focus on essentials when analyzing 12 potential factors. Suppose the dependent variable is $y = (y_1, \ldots, y_n)^T$ and the independent variable is $X = (x_{1j}, \ldots, x_{nj})^T$, where $j = 1, \ldots, p$, $\beta = (\beta_1, \ldots, \beta_p)^T$ is the coefficient vector, and the following linear model is considered:

$$y = \beta X + \epsilon. \tag{3}$$

The variable selection and parameter estimation of LASSO are obtained by Eq (4):

$$\hat{\beta} = \underset{\beta}{\arg\min} \sum_{i=1}^{n} \left( y_i - \sum_{j=1}^{p} \beta_j x_{ij} \right)^2 + \lambda \sum_{j=1}^{p} |\beta_j|. \tag{4}$$

The solution of Eq (4) can be transformed into the following optimization problem with penalty term, where $\lambda$ is the adjustment parameter, and $\lambda$ corresponding to:

$$\hat{\beta} = \underset{\beta}{\arg\min} \sum_{i=1}^{n} \left( y_i - \sum_{j=1}^{p} \beta_j x_{ij} \right)^2, \quad \text{s.t.} \quad \lambda \sum_{j=1}^{p} |\beta_j| \leq t. \tag{5}$$

**Copula**

Copula analysis consists of two steps: determining marginal distribution and analyzing dependency structure. Traditional time series models, such as ARMA and GARCH family models, are used for fitting marginal distributions. The Copula function is used to construct tail dependency.

**Marginal distribution.** In this paper, the ARMA-EGARCH model with the skewd Student-t, Student-t, generalized error and normal distributions are employed to model the marginals to capture the asymmetry and leverage effects in the

recombined series. For a stationary time series $r_t$, an ARMA(p, q) is applied to model the conditional mean model. It is represented by Eq (6).

$$r_t = c + \sum_{i=1}^{p} \theta_i r_{t-i} + \epsilon_t + \sum_{j=1}^{q} \phi_j \epsilon_{t-j},$$

(6)

where $c$ is the constant term, $\theta_i$, and $\phi_j$ are autoregressive parameters. $\epsilon_t$ is the random error of $t$ moment. $p$ and $q$ indicate the lag orders which can be determined by AIC value. The restricted variance is modeled with the EGARCH method through the following equations:

$$lnh_t = \omega + \sum_{k=1}^{\infty} \beta_k g(\xi_{t-k}) + \sum_{j=1}^{q} \alpha_j lnh_{t-j},$$

(7)

$$\epsilon_t = h_t^{1/2} \xi_t$$

(8)

$$g(\xi_t) = \theta \xi_t + \gamma(|\xi_t| - E[|\xi_t|]),$$

(9)

where $\{\alpha_j\}_{j=1,2,\ldots,q}$ and $\{\beta_k\}_{k=1,2,\ldots,\infty}$ are a nonrandom real scalar sequence, and $E_{t-1}[g(\xi_t)] = 0$. It can be seen that when $\theta < 0$, under the same fluctuation size, the increase in future conditional variance under negative fluctuations is greater than that under positive fluctuations, reflecting asymmetry and facilitating the description of financial price fluctuations. The EGARCH process not only ensures the nonnegativity of $h_t$, but also eliminates the nonnegative restriction on the coefficients of the GARCH process, and can reflect the asymmetry of volatility.

**Static and time-varying Copulas.** A copula function can connect multiple distribution functions with their respective marginal distribution functions. In recent years, many researchers have devoted themselves to extending and applying the copula method in shipping markets [13,17,42–44]. It has been suggested by Sklar [45] that any joint distribution function $F(x_1, x_2)$ with the marginal distribution of $F(x_1, x_2)$ could be represented as:

$$F(x_1, x_2) = C(F_1(x_1), F_2(x_2)).$$

(10)

If $F_1(x_1)$ and $F_2(x_2)$ are continuous and the joint distribution function is given, then

$$C(u_1, v_1) = F(F_1^{-1}(u_1), F_2^{-1}(u_2)),$$

(11)

where $u_1 = F_1(x_1), u_2 = F_2(x_2)$.

This study considers different static copula specification, and then according to the best fitted static copula specification, this study further examines the tail-dependence dynamically with corresponding time-varying copula models. In building high dependence among variables this study considers the copulas which examine the asymmetric and symmetric. The four copula functions are: Gaussian and Student-t (elliptical copulas), Clayton and Rotated Clayton (Archimedean copulas). The details can be found in Bai [13]. AIC is usually applied to determine which copula specification is the best-fitted one. So that the features of the tail dependence between the two studied variables are obtained.

In a short summary, this study's research process is shown in Fig 1. In the initial phase, the MEMD-based methodology is employed to decompose and structurally reconfigure the return series into distinct subsets representing short-term to medium- and long-term components. Subsequently, LASSO is utilized for the purpose of identifying pivotal drivers

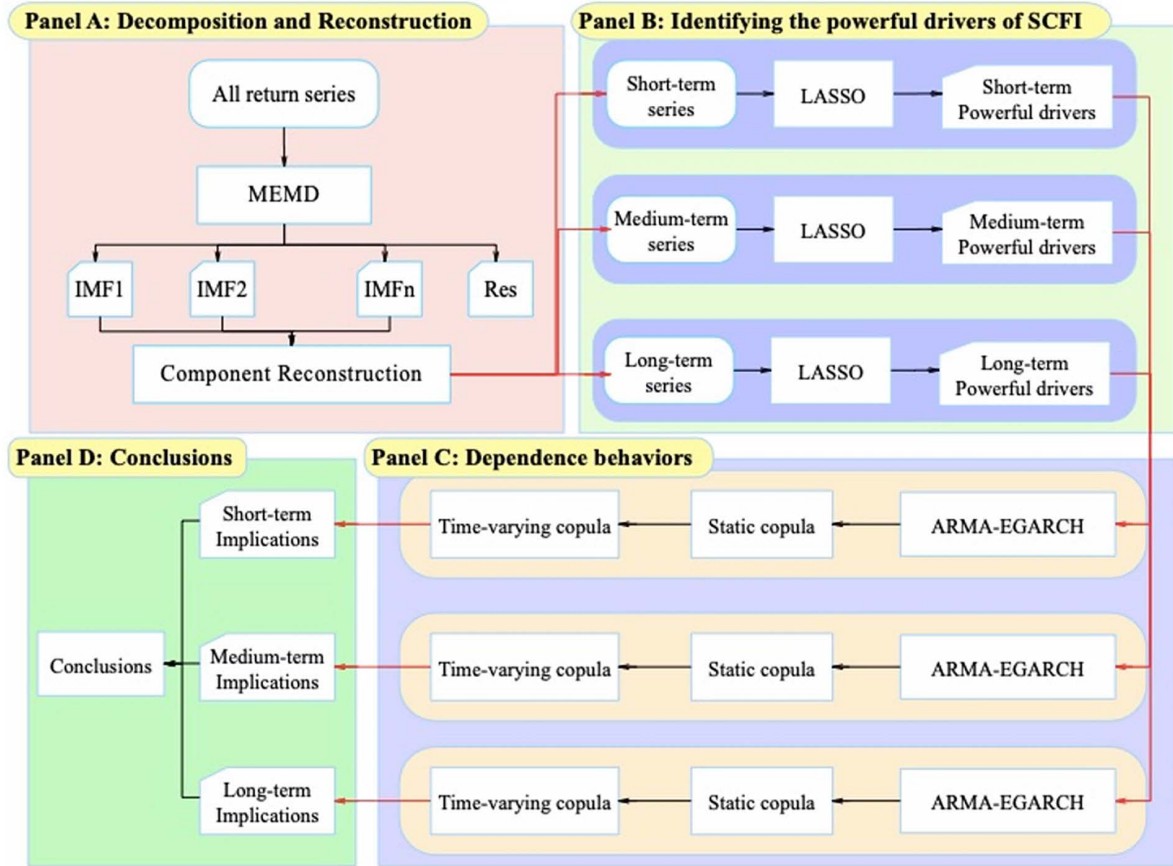

**Fig 1. The flowchart of the MEMD-based copula analysis.**

influencing SCFI. Following this, the third stage involves investigating the tail dependence relationships between SCFI and these identified influential factors using Copula models. Finally, comprehensive insights and conclusions are derived based on the findings of the analysis.

## Data

The analysis of SCFI has become increasingly important as its financial attributes have been improved. Existing literature has predominantly focused on forecasting SCFI [28–33], few study comprehensively explore the influencing factors of SCFI. To address this research gap, this paper has selected 12 different variables that presented four markets and could impact SCFI as introduced in Table 1. BDI and BDTI are from shipping market, since they belong to the same industry as SCFI and will be affected by industry policies and shipping cycles. SHCI, SP500, and MSCI_E are from stock market, since the stock market is a barometer of the economy, and during periods of economic prosperity, demand is strong and trade volume increases. Then freight rates will also rise accordingly. The US dollar is an important settlement currency in the shipping industry and also an important safe haven asset. When regional wars outbreak, the demand for the US dollar in the market will increase. Gold has always been an important safe haven asset, even against US dollar and whether gold can also hedge the risk of SCFI is the question this article aims to explore. Metal, SP _Agri, and Oil represent the commodity market. They are important

**Table 1. Selection of variables.**

| Variable | Meaning | Market | Source |
|---|---|---|---|
| SCFI | Return rate of SCFI Shanghai-Europe Container Freight Rate | Container shipping market | Clarkson |
| BDI | Return rate of Baltic Dry Freight Index | Dry bulk shipping market | Clarkson |
| BDTI | Return rate of Baltic Dirty Tanker Freight Index | Global crude oil shipping market | Clarkson |
| SP500 | Return rate of S&P 500 Index | American stock market | Wind |
| SHCI | Return rate of Shanghai Composite Index | Chinese stock market | Wind |
| MSCI_E | Return rate of Morgan Stanley Capital International Europe Index | European stock market | Wind |
| USD | Return rate of U.S. Dollar Index | Exchange market | Wind |
| Gold | Return rate of London Spot Gold Price | Gold market | Wind |
| Metal | Return rate of the Metal Component of the CRB Spot Index | Global commodity market of | Wind |
| SP_Agri | Return rate of S&P Goldman Sachs Agriculture Index | Agriculture product market | Wind |
| Oil | Return rate of West Texas Intermediate crude oil nearby future contract | Crude oil market | Wind |
| EUA | Return rate of European Union carbon emission allowance future | Carbon market | Wind |
| GPRD | Change rate of Global Geopolitical Risk Index | Geopolitical risk on a global scale | policyuncer tainty.com |

industrial raw materials. The quality of their markets is a reflection of its macroeconomic supply side. Moreover, bunker cost is an important cost in ship operation, closely related to crude oil prices. In early 2024, Europe included the shipping industry in carbon trading, strengthening the relationship between freight rates and carbon prices. Geopolitics risk has an impact on container transportation by affecting waterway safety and trade relations. The weekly data ranges from November 6th 2009 to April 25th 2025, with a total of 808 groups of observations. The original series are transformed into logarithmic returns so that findings can be compared.

Fig 2 illustrates the developmental trajectory of the return series, clearly depicting the characteristic phenomenon of volatility clustering. According to the results of Jarque-Bera test, none of the return series follows a normal distribution. Furthermore, the Augmented Dicky Fuller (ADF) [46] test indicates that all return series are stationary at the 1% significance level. The return series also have significant autocorrelation and GARCH effects, thereby it is reasonable to implement the ARMA-EGARCH model with different distributions to effectively capture the timeseries' characteristics such as volatility clustering, asymmetry, and fat-tails.

## Results

### Decomposition and reconstruction

In this paper, SCFI and potential influence drivers are decomposed into 11 groups of IMFs and one group of residuals through MEMD. Considering the complexity of the data, the permutation entropy proposed by Bandt and Pompe [47] is applied in component reconstruction. The calculation of permutation entropy can be found in Bandt and Pompe [47] and Chen et al. [48]. This paper refers to Zheng et al. [49] and set $\theta$ equal to 0.5. However, one threshold can only clarify the IMF components into two timescales, which may overlook pivotal time scales and lack accuracy. Moreover, our dataset contains data from over 15 years, so it is necessary to divide the data into different timescales. According to the permutation entropy values displayed in Table 2, the permutation entropy values of all the first four IMFs are larger than 0.5. The first four IMFs can construct a short-term timescale (from 5.18 to 7.03 weeks). Then, it can be found that the permutation entropy values of all variables' IMF7, IMF8, IMF9, IMF10, and IMF11 are almost within the scope of 0.2 to 0.3. So, this paper sets 0.3 as another threshold to divide the last six IMF components into two timescales, namely medium-term (from 29.84 to 35.35 weeks) and long-term (from 99.99 to 178.65 weeks). The periods are consistent with the findings of Chen et al. [49] in analyzing the cycles of CCFI with EMD.

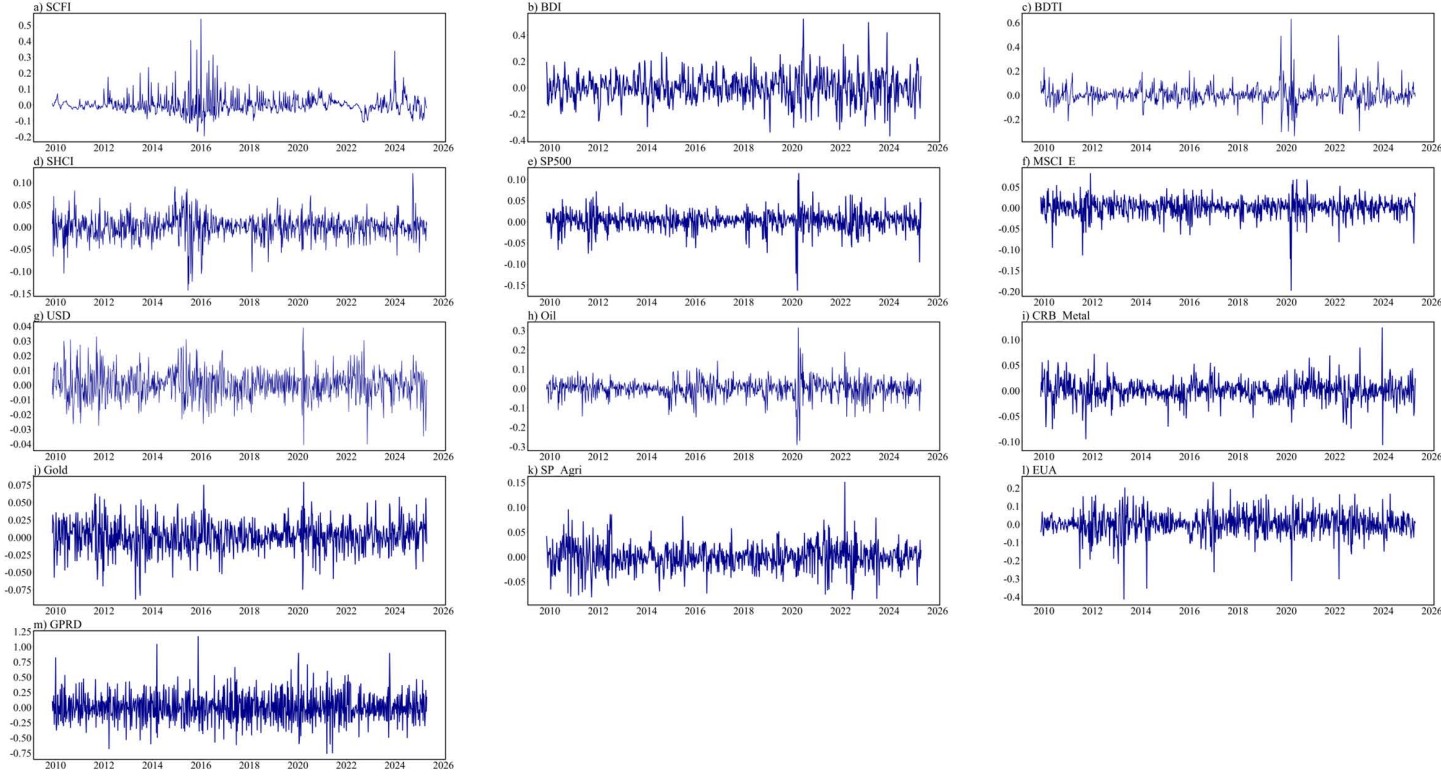

**Fig 2. Price development for all returns.**

**Table 2. Permutation entropy of each variable's IMFs.**

| – | SCFI | BDI | BDTI | SHCI | SP500 | MSCI_E | USD | Oil | Metal | Gold | SP_Agri | EUA | GPRD |
|---|------|-----|------|------|-------|--------|-----|-----|-------|------|---------|-----|------|
| IMF1 | 0.9373 | 0.9501 | 0.9628 | 0.9354 | 0.9375 | 0.9310 | 0.9457 | 0.9395 | 0.9345 | 0.9448 | 0.9402 | 0.9391 | 0.9455 |
| IMF2 | 0.8455 | 0.8454 | 0.8287 | 0.8550 | 0.8531 | 0.8395 | 0.8593 | 0.8417 | 0.8288 | 0.8203 | 0.8657 | 0.8370 | 0.8361 |
| IMF3 | 0.6809 | 0.6628 | 0.6724 | 0.6779 | 0.6779 | 0.7012 | 0.6832 | 0.6856 | 0.7022 | 0.6833 | 0.6780 | 0.6881 | 0.6933 |
| IMF4 | 0.5372 | 0.5348 | 0.5165 | 0.5317 | 0.5325 | 0.5316 | 0.5462 | 0.5372 | 0.5245 | 0.5203 | 0.5289 | 0.5251 | 0.5291 |
| IMF5 | 0.4258 | 0.4200 | 0.4266 | 0.4338 | 0.4308 | 0.4209 | 0.4326 | 0.4127 | 0.4375 | 0.4205 | 0.4217 | 0.4244 | 0.4318 |
| IMF6 | 0.3549 | 0.3427 | 0.3366 | 0.3340 | 0.3525 | 0.3475 | 0.3463 | 0.3614 | 0.3356 | 0.3398 | 0.3398 | 0.3458 | 0.3379 |
| IMF7 | 0.2877 | 0.2990 | 0.2979 | 0.2940 | 0.2952 | 0.3001 | 0.2897 | 0.3047 | 0.2926 | 0.3005 | 0.3049 | 0.2967 | 0.2981 |
| IMF8 | 0.2727 | 0.2593 | 0.2688 | 0.2654 | 0.2627 | 0.2706 | 0.2701 | 0.2676 | 0.2598 | 0.2631 | 0.2687 | 0.2691 | 0.2649 |
| IMF9 | 0.2527 | 0.2525 | 0.2479 | 0.2488 | 0.2524 | 0.2521 | 0.2386 | 0.2465 | 0.2485 | 0.2510 | 0.2386 | 0.2518 | 0.2487 |
| IMF10 | 0.2247 | 0.2431 | 0.2351 | 0.2226 | 0.2334 | 0.2347 | 0.2345 | 0.2110 | 0.2030 | 0.2364 | 0.2256 | 0.2211 | 0.2389 |
| IMF11 | 0.2262 | 0.2280 | 0.2283 | 0.2290 | 0.2293 | 0.2205 | 0.2232 | 0.2259 | 0.2260 | 0.2252 | 0.2282 | 0.2268 | 0.2320 |

The plots for different timescales of different variables are displayed in Figs A-1–A-3 in S1 Appendix, respectively. It becomes evident that the short-term sequence group exhibits numerous small and irregular fluctuations. These short-term sequences maintain the same structural characteristics as the original data, as they accurately capture the extracted short-term fluctuations. Consequently, the curves representing the medium- and long-term sequence groups display a smooth trend, devoid of such rapid fluctuations. It should be noticed that all timescale components are stationary.

## Estimation results of LASSO

LASSO, a variable selection and parameter estimation method, is chosen in this paper to effectively screen and estimate the parameters of the selected variables, considering short-, medium-, and long-term perspectives. Table 3 shows the results of LASSO estimation.

LASSO analysis reveals that developments in other shipping sectors notably influence container freight rates. The dry bulk market, in particular, has a stronger and more consistent impact on the SCFI than the tanker market. The BDI affects the SCFI as a full-frequency band (short-medium to long-term), as its market structure and demand fluctuations are closely related to container transport. In contrast, the tanker shipping market's short-term impact is limited due to differing demand structures, though long-term fluctuations can still indirectly influence the SCFI through competition for shipping resources. In the short- and medium- term, the "siphon effect" of capacity resources is manifested in the negative relationship between the rising BDI and the falling SCFI; a prolonged rise in BDI tends to reflect strong global demand for commodities, which usually corresponds to periods of global economic recovery or growth. In this context, foreign trade is also synchronized with the growth, which in turn promotes the rise in demand for container transport, and SCFI rise correspondingly. Therefore, in the long run, the BDI has a positive impact on the SCFI.

The second finding pertains to the stock market impact on container transport. In the short-term, the parameters associated with the SHCI show significance. This implies that fluctuations in the Chinese stock market can influence container freight rates in the short-term. In the medium-term, LASSO keeps both SP500 and SHCI, suggesting that not only the Chinese but also the US stock market impacts container transportation. However, MSCI_E is selected only in the long-term. Due to the close relationship between the shipping market and the economic situation, the US stock market is relatively more eye-catching compared to the European stock market, and the US stock market is still the core indicator of the world economy. In the long-term, the parameters associated with the three stock markets exhibit significance. This indicates that stock markets play a part in shaping container freight rates over the long term. Short-term stock market volatility mainly reflects changes in capital market sentiment, and its impact on SCFI is relatively indirect and limited. In the container shipping industry, especially in China as a major global manufacturing and exporting country, stock market volatility usually reflects changes in the health of the Chinese economy and foreign trade demand. European stock markets, which represent a part of the global shipping demand structure, affect the container transportation market mainly through their long-term indirect linkages with the global economy. As the stock market of one of the world's major economies, the long-term

**Table 3. Selection of variables.**

| Variable | Short-term | Medium-term | Long-term |
|---|---|---|---|
| BDI | −0.015072 | −0.161051 | 0.496760 |
| BDTI | 0 | −0.009029 | −0.012774 |
| SHCI | −0.100354 | −0.170631 | −0.442219 |
| SP500 | 0 | 0.261135 | 0.732811 |
| MSCI_E | 0 | 0 | −0.147518 |
| USD | −0.012159 | −0.048284 | 0.581249 |
| Oil | 0.010110 | −0.079312 | 0 |
| Metal | 0.037523 | 0 | 0 |
| Gold | 0 | −0.031483 | 0.302818 |
| SP_Agri | 0.008296 | −0.070384 | 0.031130 |
| EUA | −0.026401 | 0.121514 | 0.041144 |
| GPRD | −0.006708 | −0.152332 | 0.121948 |
| Intercept | 0 | 0 | 0 |

performance of the U.S. stock market directly reflects global economic growth, consumer demand, investment levels, and international trade activity. The rise or fall of the United States stock market in the medium- to long-term is often accompanied by a boom or bust in the global economy, which in turn affects the level of demand for container transportation and freight rates.

The impact of the crude oil market on the container transport market is mainly reflected in the short- and medium-term, the former is reflected in the direct impact of fuel costs, the latter indirectly through the strategic adjustment of shipping companies and policy response to the level of freight, and in the long-term its role will be gradually weakened by technological progress and structural changes.

The impact of the metal market is more important mainly in the short-term; however, the impact of the gold market is more important in the medium- and long-term. In the short-term, the metal market directly affects container demand through trade volume fluctuations. The gold market, as an indicator of macro-risk and global economic expectations, influences freight movements through investment confidence, policy environment, and other medium- to long-term paths.

Lastly, the geopolitical risks, agricultural products, carbon finance, and foreign exchange markets show enduring impacts on the container transport market from the short-term to the long-term. As global environmental awareness grows, fluctuations in the carbon market, especially in the context of carbon emission taxes and green shipping policies, may affect container freight rates by increasing or decreasing transportation costs. Geopolitical risks such as trade disputes, regional conflicts, and other factors may have an impact on the demand for container transport by disrupting the global supply chain and increasing transportation costs, etc. Against the backdrop of agricultural products being the main export cargo, price fluctuations in the agricultural products market have a direct impact on the demand for container transport. Geopolitics, agricultural markets, carbon finance and foreign exchange markets, and other factors will not only trigger short-term fluctuations in freight rates, but also through the global supply chain layout, shipping cost structure and international trade pattern of long-term shaping, in a number of points in time continue to affect the container transport market.

These conclusions provide key drivers for the container transport industry in the short-, medium- as well as long-term, especially in the face of a large number of potential predictors.

## Estimation results of copula models

This section further conducts a dependency structural analysis on the important factors identified through LASSO screening at different time intervals. The estimation results for the ARMA-EGARCH specifications are presented in Table B1 in S1 Appendix. The residual diagnosis test confirms the adequacy of the selected marginal distribution models. The estimated static copula results for different time horizons are presented in Table 4. The optimal copula is identified by cross-validation. As depicted in Figs 3–5, the time-varying nature of dynamic equivalent correlations between SCFI and selected powerful drivers can be observed across different time scales. To enhance the reliability of our results, this study excludes the initial 10% time-varying parameters from each group, and the optimal copula is identified using AIC, BIC, and maximum likelihood approaches (See Appendix C in S1 Appendix).

The initial findings reveal that no tail dependence or asymmetric tail dependence is observed in the short-term (as shown in Fig 5). The short-term dependence between factors and SCFI is limited and can be most accurately characterized by a Rotated Clayton or Gaussian distribution. Only Metal-SCFI and SP_Agri-SCFI pairs show significant results. These findings indicate that short-term SCFI is influenced by factors that either have no or higher tail dependence with them. Consequently, it is worth mentioning that upper extreme events in metal and agricultural markets could weigh on the SCFI in the short-term, pushing it higher. Because the rise in prices of these two commodities will trigger investors' concerns about inflation, the market panic caused by rising costs will lead to a joint increase in SCFI in the short-term.

Based on the short-term, no tail dependence exists in EUA-SCFI, GPRD-SCFI. Upper tail dependence exists in BDI-SCFI, SHCI-SCFI, USD-SCFI, Oil-SCFI, Metal-SCFI, and SP_Agri-SCFI. Based on the medium-term, symmetric

**Table 4. Constant copula estimation results.**

| – | BDI | BDTI | SHCI | SP500 | MSCI_E | USD | Oil | Metal | Gold | SP_Agri | EUA | GPRD |
|---|---|---|---|---|---|---|---|---|---|---|---|---|
| **Panel A: Short-term** | | | | | | | | | | | | |
| family | Rotated Clayton | – | Rotated Clayton | – | – | Rotated Clayton | Rotated Clayton | Rotated Clayton | – | Rotated Clayton | Gaussian | Gaussian |
| Par | −0.02163 | – | 0.0561 | – | – | 0.0163 | 0.0565 | 0.09107** | – | 0.0874** | 0.0046 | 0.0009 |
| LL | 0.8928 | – | 1.1460 | – | – | 0.0973 | 1.1155 | 2.7242 | – | 2.5421 | 0.0083 | 0.0003 |
| AIC | 0.2144 | – | −0.2921 | – | – | 1.8053 | −0.2310 | −3.4484 | – | −3.0844 | 1.9834 | 1.9994 |
| **Panel B: Medium-term** | | | | | | | | | | | | |
| family | Gaussian | Student-t | Student-t | Rotated Clayton | – | Clayton | Gaussian | – | Rotated Clayton | Clayton | Gaussian | Rotated Clayton |
| Par | −0.1625*** | −0.1623*** | −0.0737* | 0.0195 | – | 0.0658* | 0.0292 | – | 0.1413*** | −0.0452 | 0.0527 | 0.0623 |
| $\nu$ | – | – | 7.9878** | – | – | – | – | – | – | – | – | – |
| LL | 10.3545 | 10.3402 | 8.7151 | 0.1380 | – | 1.4281 | 0.3301 | – | 5.5878 | 1.73 | 1.0844 | 0.7317 |
| AIC | −18.7090 | −16.6805 | −13.4302 | 1.7239 | – | −0.8562 | 1.3399 | – | −9.1756 | − 1.45 | −0.1689 | 0.5365 |
| **Panel C: Long-term** | | | | | | | | | | | | |
| family | Rotated Clayton 90 degree | Rotated Clayton | Student-t | Student-t | Student-t | Rotated Clayton 90 Degree | – | – | Clayton | Student-t | Rotated Clayton | Student-t |
| Par | −0.497*** | 0.0705* | 0.0436 | −0.1308*** | −0.0577 | −0.275*** | – | – | 0.1118** | −0.0284 | −0.1224*** | −0.1154*** |
| $\nu$ | – | – | – | 8.8936*** | 7.4341 | 0.0617 | – | – | – | 7.3022 | – | 8.6668*** |
| LL | 58.6 | 1.7627 | 4.3571 | 7.2307 | 5.8858 | 18.7 | – | – | 3.3050 | 4.1992 | 6.7278 | 7.0257 |
| AIC | −115 | −1.5254 | −4.7142 | −10.4614 | −7.7715 | −35.5 | – | – | −4.6099 | −4.3985 | −11.4556 | −10.0513 |

Note: Par is the estimated parameter of the corresponding type of copula, and family is the type of optimal Copula. ***, ** and * denote the significant levels are 1%, 5% and 10% respectively.

tail dependence emerges in BDTI-SCFI and SHCI-SCFI(as shown in Table 4). This highlights the necessity for stakeholders to recognize that this correlation extends beyond the conventional linear relationship typically observed during normal market conditions. The expansion of manufacturing industry drives the synchronous growth of crude oil demand (BDTI increase) and industrial product exports (SCFI increase). A large part of the Chinese economy is driven by foreign trade, and the Chinese stock market is closely related to SCFI. The short-term market dependence of SCFI and SHCI may not be significant due to market disturbances, but they show a significant synchronous upward and downward trend in the medium-term. In the medium-term, USD-SCFI indicates lower tail dependence, because the US dollar index declines and trade demand decreases, when the expectation of a global economic recession intensifies. But in the long-term USD-SCFI indicates upper tail dependence, which indicate the public's response to economic downturns is more rapid, while their response to economic upswings is relatively slower. Gold is a widely recognized safe haven asset, and its trend often contradicts the USD index. So the tail dependence between Gold and SCFI is exactly opposite to that between SCFI and USD in the medium- and long-term. The medium-term time-varying copula results show that the dependence in extreme risk events is more powerful than that in normal periods, especially in 2012, 2016, 2020, 2022 and 2024 (as shown in Fig 4). These years were closely associated with major events that affect the economy and politics, i.e., the European debt crisis and the U.S. fiscal crisis in 2012; Brexit in 2016; the COVID-19 outbreak in 2020, geopolitical conflict in 2022, and 2024.

Moreover, the long-term dependence is more powerful than the short- and medium-term dependence(as shown in Fig 5). Overall, the results demonstrate that there is an average upper tail dependence between the carbon finance market and the container shipping market in the long-term, which is consistent with the recent research by Meng et al. [7]. An upper tail

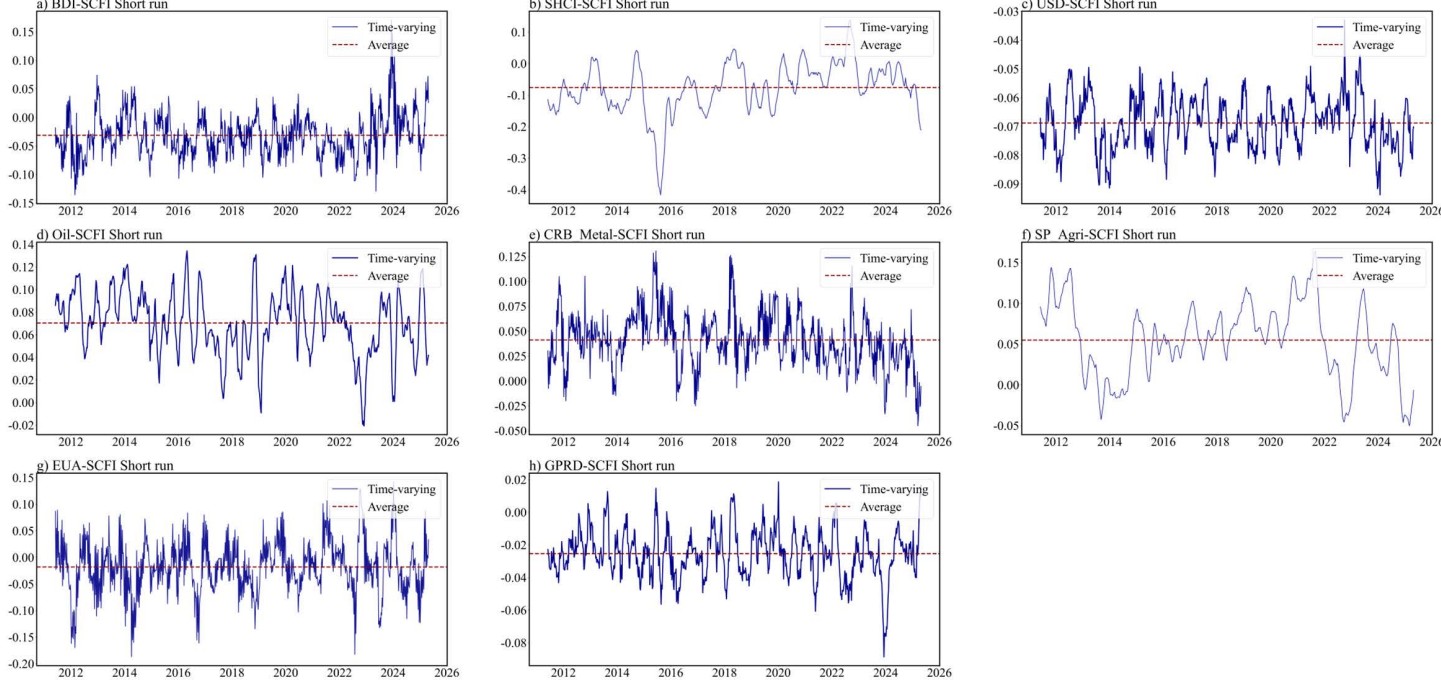

**Fig 3. Time-varying dependence parameters in the short run.**

dependence can also be observed between the bulker/tanker shipping markets and the container shipping market in the long-term. In the long-term, due to the impact of shipping cycles, the three freight rate indices tend to show an upward trend. In the long-term, the global influence of the US stock market is also reflected, with SP500-SCFI showing systematic tail dependence. The influence of European stock markets is limited. There is an upper tail dependence between Carbon price (EUA) and SCFI, as shipping companies currently use surcharges to compensate for the incremental cost caused by the shipping industry's participation in carbon trading. However, it is commonly recognized that it is easy to raise prices, but difficult to lower them. This highlights the necessity for stakeholders to recognize that the extreme upside risk in the associated shipping market, exchange market, and carbon finance market deserves to be flagged.

The dependence between SHCI and SCFI is best represented by the Student-t copula in medium-term (as shown in Table 4). Comparable patterns can be observed between SP500 and SCFI in the long-term. These results suggest there is symmetric tail dependence between the Chinese/American stock markets and the container shipping market at longer timescales. This is further supported by similar patterns observed in the long-term GPRD-SCFI pair. The dependence structure suggests that a surge in global geopolitical risks would lead to a corresponding increase in the SCFI. Geopolitical crises can affect waterway safety and trade relation, subsequently freight rates changes. Considering the market lag, so symmetrical tail dependence exists in the long-term. From the time-varying Figs 3 to 5, it is evident that the dependency structure between these two variables is relatively weak in the short-term. However, in 2022, with the escalation of the Ukraine-Russia conflict, there was an improvement observed in the long-term dependency structure between GPRD and SCFI. This crisis triggers changes in trade flows (such as economic sanctions, trade relations), which in turn lead to the reallocation of shipping capacity. Capacity adjustment is not a task that can be accomplished overnight, and Russia's influence on the global situation is important. So on the dynamic graph, an improvement is observed. The outbreak of the Syrian War in 2011 and the tense situation in the Middle East from 2016 to 2017 posed a threat to the passage of container ships in the surrounding waters, so improvements are also observed. Subsequently, the ongoing geopolitical conflict has further enhanced the long-term dependency structure.

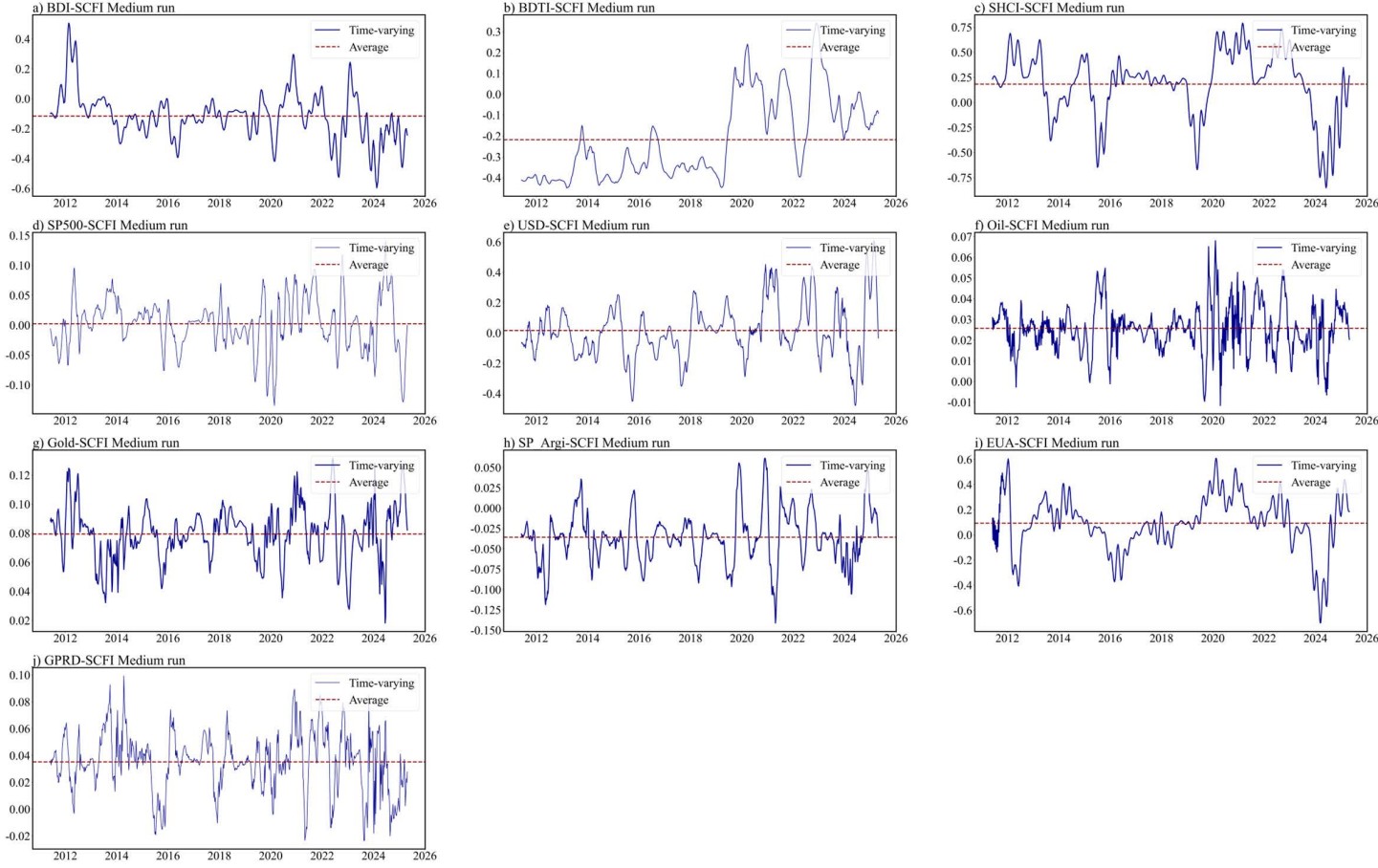

**Fig 4. Time-varying dependence parameters in the medium run.**

When assessing the relationship between the gold market and the container shipping market, it is found that the Rotated Clayton copula is best fitted for describing the tail dependence in the medium-term, while the Clayton copula is more suitable for the long-term (as shown in Table 4. This suggests that participants in container transport market need to be concerned about both the medium-term extreme upside risk and the long-term extreme downside risk to the gold market.

## Implications

The findings of this paper are useful for various stakeholders in the shipping industry and financial markets. Specifically, the whole-time horizon fluctuations in various markets, such as the dry bulk transport market, the Chinese stock market, the foreign exchange market, the agricultural market, the carbon market, and geopolitical risk, can offer valuable insights for assessing the trajectory of SCFI. In addition, short-term speculators need to pay attention to the impact of the crude oil and metals markets; medium- to long-term investors also need to pay attention to the U.S. stock market and gold market.

When engaging in tail risk management and control activities, stakeholders holding Shanghai Containerized Freight Index (SCFI) assets should particularly focus on the potential upper tail risks stemming from short-term metal and agricultural markets. In the medium-term, stakeholders should particularly focus on the potential upper tail risks stemming from gold and foreign exchange markets. Unlike the short-term, in the medium-term, risk managers in the container shipping

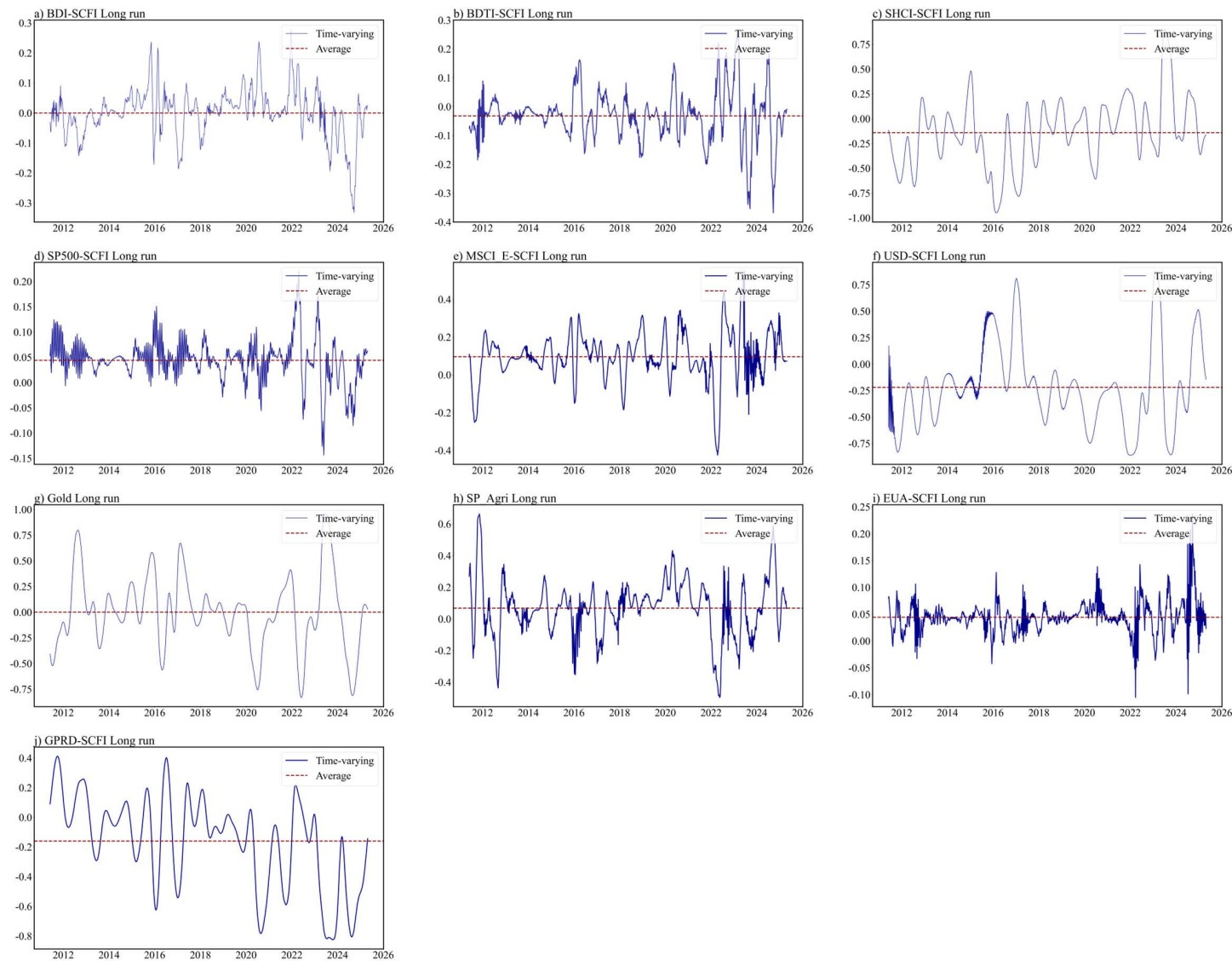

**Fig 5. Time-varying dependence parameters in the long run.**

market should be cautious of both upper and lower tail risks resulting from extreme fluctuations in the tanker transport market and the Chinese stock market. In the long-term, risk managers in the container shipping market should be cautious of upper tail risks from dry bulk transport, foreign exchange market, and carbon market, as well as lower tail risks from the gold market. Particular vigilance should be exercised against the upward and downward tail risks posed by the U.S. stock market and geopolitical risk factors. Timely precautions should be taken to avoid the risks associated with extreme volatility.

Besides, the multiscale analysis offers different implications to the stakeholders who are working in different sectors of the shipping market. Market participants can use multi-scale analysis to identify risk patterns at different timescales and adopt flexible risk response strategies. Investors need to closely follow the developments in the above markets and anticipate risks, especially in the major political and economic events on a global scale. These risks can be mitigated, for example, by

hedging strategies, building a diversified portfolio, or using derivative financial instruments. In addition, risk managers in the container transport market should ensure that they have emergency response mechanisms in place to adjust their strategies promptly in extreme risk events to minimize potential losses. The conclusion of this article is also helpful for traders of other financial assets to consider Shanghai Containerized Freight Index future when constructing investment portfolios.

## Concluding remarks

This paper distinguishes the varying influence factors of SCFI at different timescales, which supplements the research on shipping derivatives trading. To perform this goal, this paper used a MEMD-Entropy based method to decompose the multiple time series under an integrated mode and construct them into three timescales, then used LASSO to select key influencing variables under different timescales, and finally used copula to model the time-varying, asymmetric, and tail dependence. The academic contributions are as follows. Firstly, this study represents an initial endeavor to explore the relationship between container freight rates and key drivers across different periods and diverse market conditions. Secondly, the application of the MEMD method allows for the analysis of internal interactions among different drivers within our research. Lastly, the outcomes of this research hold important policy implications and strategic insights for policymakers and stakeholders in the maritime industry. The practical contribution is the insights on the risk drivers of a new tradable asset- Shanghai Containerized Freight Index (SCFI) futures. This study is valuable for entities involved in the seaborne transportation sector, including participants in spot and time-charter markets with varying time horizons of interest.

The above conclusion can be summarized as the following three points. Firstly, short-term SCFI is affected by factors that exhibit either no or upper tail dependence. Especially, upper tail extreme events from metal and agricultural markets can exert an influence on the short-term SCFI. Secondly, there is significantly greater dependence and more interdependent structure between SCFI and impact factors in the medium- and long-term. Symmetrical tail dependencies emerge in the longer time horizons. This means that lower tail extreme events can have an important impact on SCFI in both the medium- and long-term, while upper extreme events can affect it across all timeframes. Lastly, the medium- and long-term time-varying copula results show that the dependence in extreme risk events is stronger than that in normal periods, especially in 2012, 2016, 2020, 2022, and 2024. These years were closely related to major economic or political events, i.e., the European debt crisis and the U.S. fiscal crisis in 2012; Brexit in 2016; the COVID-19 outbreak in 2020, geopolitical conflict in 2022, and 2024. In the future, it is necessary and practicable to construct proper investment portfolios with different assets to manage the tail risk of SCFI future prices in different time horizons. The stakeholders can also consider SCFI future to hedge risk in commodity and financial market.

Finally, it has to admit that this article only considers the most mature and commonly used copula models. In the future, with the emergence of new mature copula models, it will inevitably bring new insights into the tail risk dependence relationship between SCFI future prices and other assets. Moreover, new methods for signal decomposition are constantly emerging, such as nonlinear mode decomposition [50]. Their application in the decomposition of shipping freight rates will further enrich the results of this research field.

## Supporting information

**S1 Data. DataSet DataEU.csv The original data used in this research.**
(CSV)

**S1 Appendix. Appendix figures and tables.**
(DOCX)

## Author contributions

**Conceptualization:** Chao Tang.

**Data curation:** Ailing Feng.

**Funding acquisition:** Yanhui Chen.

**Methodology:** Ailing Feng.

**Supervision:** Yanhui Chen.

**Writing – original draft:** Ailing Feng.

**Writing – review & editing:** Yanhui Chen.

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
