## [Decision Letter · Decision Letter 0]

15 Aug 2025

Dear Dr. Chen,

Thank you for submitting your manuscript to PLOS ONE. After careful consideration, we feel that it has merit but does not fully meet PLOS ONE’s publication criteria as it currently stands. Therefore, we invite you to submit a revised version of the manuscript that addresses the points raised during the review process.

We look forward to receiving your revised manuscript.

Kind regards,

Xu Xin

Academic Editor

PLOS ONE

Journal Requirements:

2. Thank you for stating the following in the Acknowledgments Section of your manuscript: [This research was founded by the National Natural Science Foundation of China (Grant numbers 42176217, 42476248)]

Please remove any funding-related text from the manuscript and let us know how you would like to update your Funding Statement. Currently, your Funding Statement reads as follows: [This research was founded by the National Natural Science Foundation of China (Grant numbers 42176217, 42476248). Yanhui Chen is the second recipient of both grants. The funders did not play any role in the study design, data collection and analysis, decision to publish, or preparation of the manuscript.]

3. In the online submission form, you indicated that your data is available only on request from a third party. Please note that your Data Availability Statement is currently missing the name of the contact details for the third party, such as an email address or a link to where data requests can be made. Please update your statement with the missing information.

Reviewers' comments:

Reviewer's Responses to Questions

**Comments to the Author**

1. Is the manuscript technically sound, and do the data support the conclusions?

Reviewer #1: Yes

Reviewer #2: Yes

2. Has the statistical analysis been performed appropriately and rigorously?

Reviewer #1: Yes

Reviewer #2: Yes

3. Have the authors made all data underlying the findings in their manuscript fully available?

Reviewer #1: Yes

Reviewer #2: Yes

4. Is the manuscript presented in an intelligible fashion and written in standard English?

Reviewer #1: Yes

Reviewer #2: Yes

**Reviewer #1:**

This paper takes explores the nonlinear dynamic interdependence between the containerized freight index and 12 factors from the stock, commodity, carbon, and other markets using a data decomposition–reconstruction-based time-varying copula method across multiple timescales. The findings reveal some interesting conclusions. The problem context is appealing, and the paper has some merits. However, overall, some issues still need to be improved.

1. The introduction lacks a clear focus in its literature review. Please enhance it by incorporating studies directly related to the central theme of this research.

2. Is the model robust? How does it respond to variations in the relevant parameters?

3. The paper presents abundant data; however, the analysis of the results lacks sufficient depth. Please provide a more detailed examination. Are the conclusions aligned with previous studies, or do they offer novel extensions? Please support your discussion with relevant literature.

4. What is the rationale behind the selection of variables in the paper? Please provide a deeper analysis of the trends across different variables and explain their underlying causes. Based on these trends, offer more practical management insights.

5. Please standardize the in-text citation formats across the manuscript and ensure consistency in the formatting of the reference list according to the required style guide.

6. Are there any limitations in this paper, and what areas are suggested for future research?

**Reviewer #2:**

This paper explores the nonlinear dynamic interdependence between the containerized freight index and 12 factors from the stock, commodity, carbon, and other markets using a data decomposition–reconstruction-based time-varying copula method across multiple timescales.

The topic of this article is intriguing and provides valuable insights into containerized freight rates. Generally, the paper fits with the scope of the special issue. However, several shortcomings need to be addressed in order to improve its overall quality.

Specific Comments:

1.Abstract – The abstract should clearly emphasize the core innovation and key conclusions. A thorough revision is recommended to highlight the novelty of the study and its principal findings.

2.Introduction – The introduction needs to better underscore the necessity of this research. While a literature review is provided, it does not specifically explain the rationale for selecting the particular dataset. From a methodological perspective, it lacks justification for the suitability of the chosen methods and a unique perspective on potential variables and their interrelationships.

3.MEMD and LASSO Sections – These sections mainly introduce the methods from a mathematical perspective, but lack an explanation framed from the problem-solving angle. The description should connect the methodological choices to the specific research problem and their practical implications.

4.Copula Section – In Table 4, some Copula parameters are not statistically significant (e.g., mid-term BDTI–SCFI). The paper does not discuss whether this might be due to insufficient sample size or model misspecification. It is recommended to add robustness checks to validate the results.

5.Data Sources and Variable Descriptions – The GPRD indicator is only noted by its source, without explaining its construction logic. The core variables’ calculation methods should be supplemented to verify the reasonableness of the indicators. The study uses weekly frequency data but does not justify why this is preferable to daily or monthly data—especially for short-term volatility analysis. It is suggested to provide the rationale for the choice of data frequency.

6.Depth of Result Interpretation – In Table 3, there is no explanation for why certain variables are not significant at specific scales (e.g., mid-term MSCI_E). Economic reasoning should be incorporated, such as the lagged influence of European stock markets on the Asia–Europe shipping route. In Figure 5, the description of the mechanism behind increased dependence during extreme events (e.g., the 2022 Russia–Ukraine conflict) is vague; the analysis should be linked to specific transmission channels (e.g., energy prices → capacity reallocation → freight rate volatility).

7.Conclusion – The conclusion stresses the “first-time differentiation of multiscale drivers,” but does not clarify the incremental contribution compared with existing multiscale studies. It should explicitly state how the proposed multivariate collaborative decomposition avoids scale distortion and advances the literature.

.

Reviewer #1: No

Reviewer #2: No

---

## [Author Response · Author response to Decision Letter 1]

11 Oct 2025

Responses to Reviewer #1:

Thank you very much for your insightful comments. In this version, we revised our paper according to your suggestions. In the revised manuscript, we highlight the places that have been modified. Please refer to our revised version. Point-by-point responses are listed below

1. The introduction lacks a clear focus in its literature review. Please enhance it by incorporating studies directly related to the central theme of this research.

Response: In the revised version, we have added 8 articles on shipping derivatives, since the motivation is researching the potential risk linkage between container freight rate futures and other assets which can assist some financial institutions in using SCFI futures for risk management of other assets. But previous research subjects are Forward Freight Agreements (FFAs) traded on the Baltic Exchange in associated with the dry bulk and tanker markets. There is a lack of research related to container freight rate derivatives. In addition, most of them only focus on the nexus between the spot market and FFA. Please refer to the third paragraph in Section 1.

2. Is the model robust? How does it respond to variations in the relevant parameters?

Response: The model is robust. First, during the LASSO parameter estimation, both the Akaike Information Criterion (AIC) and cross-validation methods are employed, and the results consistently select the same hyperparameter values. Second, in the copula estimation process, the optimal copula is identified using AIC, BIC, and maximum likelihood approaches, with the majority of these methods yielding consistent results. These findings collectively confirm the robustness of the proposed model. In the revised version we add the value of AIC, BIC, and maximum likelihood approaches in copula estimation. Please refer to Appendix C in the end of the article.

When we conducted the robustness check, we also found some typos, such as, in the mid-term the dependence between USD and SCFI is best represented by Clayton instead of Rotated Clayton, the parameter value was right but the name was incorrectly. Pleases refer to the Table 4.

3. The paper presents abundant data; however, the analysis of the results lacks sufficient depth. Please provide a more detailed examination. Are the conclusions aligned with previous studies, or do they offer novel extensions? Please support your discussion with relevant literature.

Response: We add detailed explanation in Section 4.3. Please refer to the highlight sentences in Paragraph 2, 3, 4, and 5.

This is the first paper about the tail risk dependence between SCFI and other assets which is benefit in exploring the risk management in associated with SCFI future. Some of the previous researches focus on the nexus between dry bulk freight rates and iron ore prices, tanker freight rates and oil prices, which are not suitable to make a comparison.

4. What is the rationale behind the selection of variables in the paper? Please provide a deeper analysis of the trends across different variables and explain their underlying causes. Based on these trends, offer more practical management insights.

Response: In this version, we explain why we chose these variables and we analyze the results from an economic perspective. Please refer to the first paragraph in Section 3 for the explanation of variable selection and the highlight sentences in Section 4.3 for economic explanation.

5. Please standardize the in-text citation formats across the manuscript and ensure consistency in the formatting of the reference list according to the required style guide.

Response: We refreshed the citation formats.

6. Are there any limitations in this paper, and what areas are suggested for future research?

Response: Yes. The results of this article are helpful for stakeholders to use different assets and SCFI futures to manage tail risks in different cycles with proper investment portfolios. However, this article does not provide specific investment portfolio recommendations. The author will explore the construction of investment portfolios more from practice in future work, using the results of this article. Moreover, this article is practice oriented, so mature copula models were used. In the future, we will consider introducing more copula models for analysis. In the revised version, we mentioned the limitations in the end of Section 5 and please refer to the highlight sentences.

Reviewer #2:

Thank you very much for your insightful comments. In this version, we revised our paper according to your suggestions. In the revised manuscript, we highlight the places that have been modified. Please refer to our revised version. Point-by-point responses are listed below.

1.Abstract – The abstract should clearly emphasize the core innovation and key conclusions. A thorough revision is recommended to highlight the novelty of the study and its principal findings.

Response: We revised the abstract according to your advice. Thank you! Briefly speaking, this paper is the first article focuses on the tail risk interdependence analysis between Shanghai Containerized Freight Index and other assets, because the launch of Shanghai Containerized Freight Index (SCFI) futures makes it inevitable for stakeholders to use it to build risk management portfolios with other assets.

2.Introduction – The introduction needs to better underscore the necessity of this research. While a literature review is provided, it does not specifically explain the rationale for selecting the particular dataset. From a methodological perspective, it lacks justification for the suitability of the chosen methods and a unique perspective on potential variables and their interrelationships.

Response: We mentioned the necessity of research in the Introduction, such as the last sentence of the first paragraph and the comments on the literature in the last sentences of Paragraph 3. And the summary of all existing literature reviews in Paragraph 4 and 5 confirms the necessity of this study. In a short summary, the necessity of research includes the following points:

(1) The rising demand for risk management in container transportation is driven by internal and external environmental factors.

(2) All the research subjects are the relations between spot freight rates and FFAs in associated with the dry bulk and tanker markets. There is a lack of research related to container freight rate derivatives.

(3) The existing researches predominantly focus on static and dynamic analysis of freight rates at the overall level.

The first point describes the objective demand. The last two points describes the limitation of previous research from the aspects of research targets and methods.

In this version, we explain the applicability of the method (please refer to Section 2.1 and 2.2) and reason of choosing these variables (please refer to the first paragraph in Section 3).

3.MEMD and LASSO Sections – These sections mainly introduce the methods from a mathematical perspective, but lack an explanation framed from the problem-solving angle. The description should connect the methodological choices to the specific research problem and their practical implications.

Response: We added some explanations in Section 2.1 and Section 2.2

4.Copula Section – In Table 4, some Copula parameters are not statistically significant (e.g., mid-term BDTI–SCFI). The paper does not discuss whether this might be due to insufficient sample size or model misspecification. It is recommended to add robustness checks to validate the results.

Response: The copula is used to determine the tail risk dependency relationship between the critical variables and SCFI in different time domain. In Table 4, some Copula parameters are not statistically significant, because this article only selected four common copula models and none of those models can measure the significant tail dependencies accurately. This is the limitation of this article, which we mentioned in the end of Section 5 and please refer to the highlight sentences.

Refer to the sample size, Marco Bee(2011) mentioned in his article about copula method research that the approximation is acceptable with sample size=500. Jan-Michael Becker(2022) indicated the minimum sample size is 800 which can make the copula term’s parameter estimation to achieve power levels of 80% and higher. The sample size in our paper is 808 (from November 6th 2009 to April 25th 2025), which guaranteed the acceptable results. Moreover, SCFI was launched in the middle of October 2009. This sample size is the maximum sample size we can obtain.

Under your suggestion, we have re-examined the issue of model robustness. First, during the LASSO parameter estimation, both the Akaike Information Criterion (AIC) and cross-validation methods are employed, and the results consistently select the same hyperparameter values. Second, in the copula estimation process, the optimal copula is identified using AIC, BIC, and maximum likelihood approaches, with the majority of these methods yielding consistent results. These findings collectively confirm the robustness of the proposed model. In the revised version we add the value of AIC, BIC, and maximum likelihood approaches in copula estimation. Please refer to Appendix C in the end of the article.

5.Data Sources and Variable Descriptions – The GPRD indicator is only noted by its source, without explaining its construction logic. The core variables’ calculation methods should be supplemented to verify the reasonableness of the indicators. The study uses weekly frequency data but does not justify why this is preferable to daily or monthly data—especially for short-term volatility analysis. It is suggested to provide the rationale for the choice of data frequency.

Response: Shanghai Containerized Freight Index is published weekly. So we have to use weekly data in this research. Container booking is usually done about 1-2 weeks in advance. It is not measured on a daily basis. Moreover, MEMD is a self-adaptive method for decomposing multivariate time series data into different frequency domain data within a unified framework. The data used in this article belongs to different markets, and the synergy between the data needs to be considered when analyzing. MEMD perfectly meets this requirement. The data frequency in this article is not artificially selected, but objectively chosen. The objective of this article is not to analyze the rise and fall cycles of various assets, so this objective method was chosen for analysis. The revised version we emphases the adaptivity of MEMD in Section 2.1 and please refer to the highlight words.

6.Depth of Result Interpretation – In Table 3, there is no explanation for why certain variables are not significant at specific scales (e.g., mid-term MSCI_E). Economic reasoning should be incorporated, such as the lagged influence of European stock markets on the Asia–Europe shipping route. In Figure 5, the description of the mechanism behind increased dependence during extreme events (e.g., the 2022 Russia–Ukraine conflict) is vague; the analysis should be linked to specific transmission channels (e.g., energy prices → capacity reallocation → freight rate volatility).

Response: In the revised version, we add the explanation on why MSCI_E is not significant in the mid-term. Please refer to the highlight sentences in Paragraph 3 Section 4.2. Moreover, using lasso for variable screening is only the first step and the research target is the tail dependency for risk management. Although Lasso selected some variables (e.g., short-term SHCI), their tail dependence with SCFI is not obvious, and the policy recommendations are based on tail dependence, so no in-depth analysis was conducted here. But in the revised version, we add detailed economic explanation on the tail dependence. Please refer to the highlight sentences in Paragraph 2, 3, 4, and 5 Section 4.3.

Refer to the explanation of long-term tail dependence between GPRD and SCFI, we specific the transmission channels (security of waterways/changes in trade relations→ capacity reallocation → freight rate volatility). Please refer to the highlight sentences (second part) in Paragraph 5 Section 4.3.

7.Conclusion – The conclusion stresses the “first-time differentiation of multiscale drivers,” but does not clarify the incremental contribution compared with existing multiscale studies. It should explicitly state how the proposed multivariate collaborative decomposition avoids scale distortion and advances the literature.

Response: This is the first paper to distinguish the varying influence factors of containerized freight rates at different timescales. The incremental contribution is not the innovation in basic method but proposing a framework to analyze the tail risk dependence between container freight rates and other assets, which supplements the research on shipping derivatives trading. In the revised version, we change the order of the contributions and please refer to the highlight sentence in Section 5.

The multivariate collaborative decomposition method used in this paper is Multivariate Empirical Mode Decomposition. It was proposed by Rehman and Mandic (2010). We are just using it to conduct multiple timescales analysis. It is a self-adaptive method for decomposing multivariate time series data into different frequency domain data within a unified framework. It has been used in similar scenarios in the past ten years, such as Sun(2014), and Zhu et al. (2019). In the revised version, we clarify the advantage in Section 2.1.

References

[1]. Bee M . Adaptive Importance Sampling for simulating copula-based distributions. Insurance Mathematics & Economics, 2011, 48(2):237-245. DOI:10.1016/j.insmatheco.2010.11.004.

[2]. Becker J M , Proksch D , Ringle C M . Revisiting Gaussian copulas to handle endogenous regressors. Journal of the Academy of Marketing Science, 2022, 50.DOI:10.1007/s11747-021-00805-y.

[3]. Rehman N, Mandic DP. Multivariate empirical mode decomposition. Proceedings of The Royal Society A, 2010, 466(2117): 1291-1302.DOI:10.1098/rspa.2009.0502.

[4]. Sun X , Tang L , Yang Y ,et al. Identifying the dynamic relationship between tanker freight rates and oil prices: In the perspective of multiscale relevance. Economic Modelling, 2014, 42:287-295.DOI:10.1016/j.econmod.2014.06.019.

[5]. Zhu B, Ye S, Han D, Wang P, He K, Wei Y-M, et al. A multiscale analysis for carbon price drivers. Energy Economics, 2019, 78:202-216.DOI:10.1016/j.eneco.2018.11.007.

---

## [Decision Letter · Decision Letter 1]

26 Nov 2025

Dear Dr. Chen,

Thank you for submitting your manuscript to PLOS ONE. After careful consideration, we feel that it has merit but does not fully meet PLOS ONE’s publication criteria as it currently stands. Therefore, we invite you to submit a revised version of the manuscript that addresses the points raised during the review process.

We look forward to receiving your revised manuscript.

Kind regards,

Feier Chen, Ph.D

Academic Editor

PLOS ONE

Journal Requirements:

Reviewers' comments:

Reviewer's Responses to Questions

**Comments to the Author**

Reviewer #2: (No Response)

Reviewer #3: All comments have been addressed

Reviewer #4: (No Response)

Reviewer #5: All comments have been addressed

2. Is the manuscript technically sound, and do the data support the conclusions?

Reviewer #2: (No Response)

Reviewer #3: Yes

Reviewer #4: Yes

Reviewer #5: Yes

3. Has the statistical analysis been performed appropriately and rigorously?

Reviewer #2: (No Response)

Reviewer #3: Yes

Reviewer #4: Yes

Reviewer #5: Yes

4. Have the authors made all data underlying the findings in their manuscript fully available?

Reviewer #2: (No Response)

Reviewer #3: Yes

Reviewer #4: Yes

Reviewer #5: Yes

5. Is the manuscript presented in an intelligible fashion and written in standard English?

Reviewer #2: (No Response)

Reviewer #3: Yes

Reviewer #4: Yes

Reviewer #5: Yes

Reviewer #2: Overall Comments:

This paper explores the nonlinear dynamic interdependence between the containerized freight index and 12 factors from the stock, commodity, carbon, and other markets using a data decomposition–reconstruction-based time-varying copula method across multiple timescales.

The paper has undergone overall revisions, but the workload still appears to be insufficient. The practical implications of the findings have not been clearly demonstrated. Additionally, the paper contains numerous informal expressions, the relevant validations of the model are incomplete, and it lacks the necessary academic rigor.

Specific Comments:

1. In the abstract, what phenomenon of the Shanghai Containerized Freight Index (SCFI) futures highlights the growing demand for risk management in the container shipping market?

2. In the abstract, the research on the factors behind container freight rate fluctuations is still limited. How is this reflected?

3. In the abstract, this paper provides important reference significance for various stakeholders with different financial attributes in activities such as trading and risk management, covering a wide time frame. This can be outlined in the abstract.

4. The article contains colloquial expressions, and I suggest proofreading the entire text.

5. The paper conducts extensive analysis from the perspective of dynamic multiscale dependencies to identify the key driving factors of container freight rates. What is the practical value of the core results in guiding practice?

6. The research contribution of this study repeatedly emphasizes that certain research is being conducted for the first time. Is this valid? Is there sufficient support for this claim? It is suggested to elaborate on this aspect.

7. Is the necessary validation of the model analysis complete? It is recommended to demonstrate the feasibility of the model from different perspectives.

Reviewer #3: The revised version shows substantial improvements in structure, clarity, and methodological justification.

Reviewer #4: The paper analyses the dependence between the Shanghai Containerized Freight Index and 12 other markets, claiming to find different dependence dynamics between different these markets at the three different timescale categories they consider.

The authors are correct that consideration of multiscale dynamics is a significant step forward for this field. Financial markets, like so many other systems, are well known to interact and evolve differently on different timescales, and so the authors’ findings that there are differences are not surprising, but their identification of what these differences are is an important contribution to the field.

The manuscript is clear and understandable and all sufficiently claims are supported. The authors have also made reasonable edits based on the previous reviews, which have improved the manuscript.

The MEMD and copula methodologies are widely used in financial modelling. Nevertheless, I encourage the authors to consider in further analysis for future contributions-

• Oscillatory-based nonlinear mode decomposition, which is well established in dynamical systems theory, and I believe has the potential for both more granular (in time) and accurate time-series decompositions (Iatsenko, McClintock, Stefanovska, Physical Review E 2015).

• That while the complexity of financial market dynamics makes their modelling as stochastic (e.g. by copula) not unreasonable, market interactions are fundamentally deterministic and application of stochastic methods only is insensitive to the underlying determinism (Rowland Adams, Newman, Stefanovska, EPSJ:ST 2023). There are methods from multiscale nonlinear systems theory that are designed for application to this kind of system, and could give highly informative results, in addition to the copula analysis (https://ui.adsabs.harvard.edu/abs/2019zndo…3470856I/abstract, User Manual).

However, I repeat that the authors’ findings are interesting and publishable in their own right, and that application of the above methods is more appropriate for a separate contribution than inclusion in this current one at this stage.

Reviewer #5: The authors have satisfactorily addressed the concerns raised during the previous round of review. The revised manuscript is significantly improved in terms of methodological robustness and the depth of economic interpretation.

I recommend a final round of language polishing. Examples:

1. “skewd student-t” → “skewed Student-t” in the EGARCH description.

2. Missing spaces around some symbols, e.g. “λis the adjustment parameter” → “λ is the adjustment parameter.”

.

Reviewer #2: No

Reviewer #3: No

Reviewer #4: No

Reviewer #5: **Yes:** Tianyi ZhengTianyi ZhengTianyi ZhengTianyi Zheng

---

## [Author Response · Author response to Decision Letter 2]

5 Jan 2026

Please refer to the attached file.

---

## [Decision Letter · Decision Letter 2]

19 Feb 2026

Identifying the significant drivers of containerized freight rates: From the perspective of dynamic multiscale dependence

PONE-D-25-32573R2

Dear Dr. Chen,

We’re pleased to inform you that your manuscript has been judged scientifically suitable for publication and will be formally accepted for publication once it meets all outstanding technical requirements.

Kind regards,

Feier Chen, Ph.D

Academic Editor

PLOS One

Additional Editor Comments (optional):

Reviewers' comments:

Reviewer's Responses to Questions

**Comments to the Author**

Reviewer #2: All comments have been addressed

2. Is the manuscript technically sound, and do the data support the conclusions?

Reviewer #2: Yes

3. Has the statistical analysis been performed appropriately and rigorously?

Reviewer #2: Yes

4. Have the authors made all data underlying the findings in their manuscript fully available?

Reviewer #2: No

5. Is the manuscript presented in an intelligible fashion and written in standard English?

Reviewer #2: Yes

Reviewer #2: The authors have thoroughly addressed all the concerns I raised in my previous review. I appreciate the revisions and clarifications made to the manuscript. I have no further questions or suggestions.

.

Reviewer #2: No

---

## [Editor Report · Acceptance letter]

PONE-D-25-32573R2

PLOS One

Dear Dr. Chen,

I'm pleased to inform you that your manuscript has been deemed suitable for publication in PLOS One. Congratulations! Your manuscript is now being handed over to our production team.

Kind regards,

on behalf of

Dr. Feier Chen

Academic Editor

PLOS One